# THINK BEYOND SIZE: ADAPTIVE PROMPTING FOR MORE EFFECTIVE REASONING

## ABSTRACT

Pretrained large language models (LLMs) are increasingly utilized across a wide range of natural language processing (NLP) tasks due to their impressive capabilities as few-shot learners. Recent techniques, such as chain-of-thought (CoT) prompting, have significantly advanced multi-step reasoning by introducing step-by-step decomposition, achieving state-of-the-art results on complex reasoning benchmarks. However, these approaches often rely on static prompting templates that do not adapt to task complexity or errors during the reasoning process. In this work, we introduce Adaptive Prompting, a dynamic and iterative framework designed to enhance reasoning by incorporating real-time adjustments to prompt structures and validation mechanisms.Experimental results demonstrate that Adaptive Prompting significantly improves performance on diverse reasoning benchmarks, including arithmetic reasoning (GSM8K, MultiArith), logical reasoning and commonsense tasks, achieving substantial accuracy gains compared to static prompting baselines. By integrating guided prompts, intermediate validation, and self-corrective steps, our approach enables smaller models to achieve competitive performance with larger counterparts, such as GPT-4, while maintaining computational efficiency. The framework achieves this without requiring fine-tuning or task-specific training data, highlighting the untapped potential of iterative reasoning methods.

## 1 INTRODUCTION

Large Language Models (LLMs) have emerged as powerful tools in natural language processing (NLP), demonstrating exceptional capabilities in tasks such as machine translation, summarization, and question answering Brown et al. (2020). Recent developments have highlighted the transformative potential of few-shot and zero-shot learning paradigms, enabling LLMs to generalize across tasks with minimal examples or additional training Kojima et al. (2022). These approaches have expanded the applicability of LLMs while reducing reliance on extensive task-specific datasets, paving the way for more generalized AI solutions.

The conventional strategy to enhance model performance has largely focused on scaling. Models like GPT-3, with 175 billion parameters Brown et al. (2020), and GPT-4, surpassing a trillion parameters OpenAI (2023), have demonstrated remarkable reasoning capabilities attributed to their capacity to capture complex linguistic patterns. However, this emphasis on scaling comes with significant drawbacks. Larger architectures impose substantial computational costs, memory overhead, and energy consumption, making them impractical for many real-world applications Tian et al. (2024). Furthermore, scaling does not guarantee proportional improvements in performance; diminishing returns are frequently observed, particularly in tasks requiring nuanced reasoning or domain-specific knowledge Zhong et al. (2024). These limitations have motivated a shift in focus from merely increasing model size to optimizing task interaction through advanced prompting techniques.

Prompting has become a critical tool for maximizing LLM performance. Chain-of-Thought (CoT) prompting, introduced by Wei et al. Wei et al. (2022), enhances reasoning by providing structured examples that guide the model through incremental reasoning steps. Similarly, zero-shot CoT prompting, which uses phrases like **"Let's think step by step"**, encourages the model to independently articulate its reasoning Kojima et al. (2022). While these methods show promise, they

often lack consistency and robustness in complex tasks. Dynamic prompting techniques, such as those proposed by Wang et al. Wang et al. (2023), represent a significant evolution, tailoring the complexity of prompts to task requirements. However, existing approaches frequently lack iterative validation or error-correction mechanisms, limiting their effectiveness in scenarios requiring complex reasoning Zhang et al. (2022).

Adaptive Prompting addresses these limitations by introducing a flexible framework that integrates iterative reasoning and systematic validation into the prompting process. Unlike static prompting strategies, Adaptive Prompting dynamically adjusts prompt structures in real time based on task complexity and model performance. It incorporates mechanisms for guided reasoning, intermediate validation, and error correction, providing a robust alternative to traditional methods. By focusing on optimizing the reasoning process itself, Adaptive Prompting enables smaller models to achieve performance levels comparable to larger counterparts, challenging the prevailing assumption that model size is the primary determinant of reasoning efficacy Srivastava et al. (2024).

This paper explores the theoretical foundations of Adaptive Prompting, evaluates its effectiveness on diverse reasoning benchmarks, and demonstrates its potential to democratize access to high-performing AI systems. By reducing dependency on large-scale computational resources, Adaptive Prompting offers a scalable solution for reasoning-intensive applications, paving the way for advancements in adaptive and explainable AI.

## 2 RELATED WORK

The performance of Large Language Models (LLMs) in reasoning tasks has been greatly influenced by various prompting techniques, which provide different methods of guiding the model's reasoning process. Among the most well-known techniques are few-shot prompting, zero-shot prompting, and dynamic prompting.

Few-shot prompting is a technique in which a small number of examples are provided alongside the query to guide the model's reasoning. This method has been popularized by the work of Brown et al. Brown et al. (2020), where they demonstrate that even without task-specific training data, models like GPT-3 can generalize well to unseen tasks. The model learns to follow the examples and applies similar reasoning to the given task. This approach allows LLMs to perform well in tasks where labeled data is scarce, making it a powerful tool for a wide range of applications. However, few-shot prompting has its limitations. The primary challenge is that the model may struggle with tasks requiring complex reasoning if the provided examples are insufficient or overly simplistic. Additionally, the model's reasoning can become inconsistent if the examples do not sufficiently capture the complexity of the problem. Some studies have shown that when tasked with multi-step or abstract reasoning, the model's output can deviate from expected results, highlighting the importance of providing well-chosen examples that cover a wide range of reasoning paths Wei et al. (2022).

Zero-shot prompting, as explored in Kojima et al. Kojima et al. (2022), involves giving the model a task without any prior examples, relying instead on a simple instruction like "Let's think step by step." This approach forces the model to perform reasoning based purely on its pre-existing knowledge, making it useful in tasks where providing examples is impractical or infeasible. Zero-shot prompting reduces the reliance on task-specific data and allows for generalized reasoning across domains. However, it has been found to lead to significant variability in reasoning, especially in complex or multi-step problems. Zero-shot reasoning can be prone to errors, as models may fail to maintain logical consistency or may misinterpret the task, resulting in less reliable answers. Models may also fail to capture intricate problem dependencies, leading to answers that appear superficially correct but fail under deeper scrutiny Wei et al. (2022).

Chain-of-thought (CoT) prompting, introduced by Wei et al. Wei et al. (2022), aims to enhance reasoning by breaking down the problem into intermediate steps, making the reasoning process more transparent and interpretable. The model is instructed to explain its thought process step-by-step, which not only improves answer accuracy but also provides insights into how the model arrived at its conclusion. This method has shown significant improvements in tasks that require detailed logical steps or multi-step problem solving. However, despite its advantages, CoT prompting often results in longer and more verbose outputs, and the model's reasoning may still be prone to errors in

multi-step tasks, particularly in cases involving abstract reasoning or complex logical dependencies. Moreover, the verbosity of the responses can increase computational overhead, which may not be ideal for all use cases, particularly when quick responses are required Wei et al. (2022).

Dynamic prompting extends traditional approaches by adapting the prompt based on the task complexity or the model's intermediate performance. Wang et al. Wang et al. (2023) propose a dynamic prompting technique that adjusts the number of in-context examples according to the complexity of the input and the available computational budget. This approach is beneficial because it allows for a flexible adaptation to different tasks, improving the model's efficiency by reducing unnecessary computations. Dynamic prompting allows for better resource management, ensuring that more computationally demanding tasks are met with more refined prompts, while simpler tasks use fewer examples. However, existing dynamic prompting methods often lack built-in mechanisms for iterative validation and error correction, which can affect their reliability when handling complex reasoning tasks. Without a mechanism to refine intermediate answers or correct erroneous reasoning, dynamic prompting, though efficient, can sometimes lead to suboptimal or incorrect outputs.

Our approach, Adaptive Prompting, builds upon the foundations of few-shot, zero-shot, and dynamic prompting. It introduces an iterative reasoning process with challenge-rebuttal stages to cross-verify the model's reasoning and refine the final answer. By focusing on refining the reasoning process itself rather than just increasing model size, Adaptive Prompting helps reduce computational resource dependencies. It does this by dynamically adjusting the number of reasoning steps, iteratively validating intermediate results, and integrating corrective feedback to ensure that the final output is both accurate and reliable. This method is designed to dynamically adjust prompts based on task complexity and performance, allowing smaller models to achieve performance levels comparable to larger counterparts. Through this process, Adaptive Prompting ensures that even in the absence of extremely large models, reasoning accuracy can be achieved with minimal computational cost, making it a promising technique for both general and complex reasoning tasks.

## 3 METHODOLOGY

This paper explores the effectiveness of Adaptive prompting in improving the accuracy of responses generated by smaller large language models (LLMs) with fewer parameters.

### 3.1 ABOUT THE DATASET

Model's performance on a range of arithmetic reasoning benchmarks, including MultiArith Kojima et al. (2022), SVAMP Patel et al. (2021), AddSub, GSM8K Cobbe et al. (2021), AQuA, and SingleEq. These datasets test skills from basic operations to complex multi-step problem-solving. For instance, MultiArith and SVAMP assess multi-step reasoning, while AddSub focuses on simple arithmetic, and GSM8K evaluates grade-school-level problem-solving. Additionally, we tested commonsense reasoning using CSQA and StrategyQA, which require the model to apply everyday knowledge and strategic thinking. This evaluation provides a comprehensive understanding of the model's ability to handle both structured mathematical problems and more open-ended reasoning tasks.

### 3.2 MODEL SELECTION

For this study, we selected the gemma2-9b-it model, which features 9 billion parameters. This model was chosen due to its balance between computational efficiency and language processing capability. While larger models like GPT-3.5, with 175 billion parameters, and GPT-4, with approximately 1.76 trillion parameters, offer substantial power, they require significant computational resources. In contrast, the gemma2-9b-it model provides faster inference times, making it well-suited for real-time applications, yet still delivers robust performance in general language tasks.

### 3.3 EVALUATION SETUP

To evaluate the effectiveness of dynamic prompting, we conducted experiments comparing its performance to traditional zero-shot prompting across the aforementioned datasets. Both smaller models like gemma2-9b-it and larger models like GPT-3.5 and GPT-4 were assessed, allowing for a

comparative analysis of the impact of model size and prompting strategy. Accuracy scores across each dataset were recorded, highlighting the ability of dynamic prompting to enhance performance, particularly in smaller models, by guiding the reasoning process more effectively than static zero-shot methods.

## 4  DESIGN AND IMPLEMENTATION

Adaptive Prompting introduces a dynamic, multi-stage reasoning framework for language models, enabling them to tackle complex, multi-step problems with greater reliability and accuracy.

Unlike static approaches, such as Chain-of-Thought (CoT) prompting, Adaptive Prompting incorporates mechanisms for iterative refinement, real-time adaptability, and error correction, allowing models to dynamically adjust their reasoning pathways.

This framework mimics human cognitive processes by focusing on understanding, validating, and refining solutions rather than solely generating output in a single pass.

To illustrate the methodology, consider a more complex mathematical word problem:

> *"A farmer starts with 24 apples. He sells one-third of them to a customer and gives half of the remaining apples to a friend. Later, he picks 18 more apples from his orchard and then sells a quarter of his total apples to another customer. How many apples does the farmer have now?"*

Traditional prompting methods, such as CoT, would instruct the model to solve the problem step-by-step, assuming that intermediate calculations are accurate. However, CoT lacks mechanisms to detect or correct errors once they occur, leading to potential propagation of inaccuracies. Adaptive Prompting addresses this limitation by introducing a multi-stage approach where the reasoning process evolves through understanding, hypothesis generation, validation, and refinement.

The process begins with a deep engagement with the problem. The model is guided to identify the key components and relationships within the question, ensuring a comprehensive understanding before performing any calculations. The problem is framed in a way that encourages the model to decompose it into manageable parts. For example, the model might be instructed:

> *"Before solving the problem, identify the quantities and operations involved. What does the problem ask us to calculate, and what steps will help us reach the solution?"*

In response, the model identifies the sequential operations: starting with 24 apples, selling one-third $(24/3)$, subtracting this amount, halving the remaining apples, adding 18 apples, and finally selling a quarter of the total. This stage, integral to **Adaptive Prompting**, ensures that the problem is fully dissected, reducing the risk of misinterpreting or oversimplifying the task.

Once the problem is understood, the model generates an initial hypothesis or solution based on its structured breakdown. For the given example, the reasoning proceeds as follows:

> *"The farmer starts with 24 apples. Selling one-third means he sells $24/3 = 8$ apples, leaving $24 - 8 = 16$ apples. Giving half of the remaining apples to a friend means he gives away $16/2 = 8$ apples, leaving $16 - 8 = 8$ apples. Next, he picks 18 apples, resulting in $8 + 18 = 26$ apples. Finally, selling a quarter of his total apples means he sells $26/4 = 6.5$ apples, leaving $26 - 6.5 = 19.5$ apples."*

In traditional approaches, this solution might be accepted as final. However, **Adaptive Prompting** treats it as a preliminary hypothesis, which must be scrutinized for logical consistency and accuracy. The intermediate solution undergoes a validation phase, where the model critically reviews each calculation. This process mirrors human self-reflection, encouraging the model to revisit and verify its steps. For instance, the model might be instructed:

> *"Review each step of your solution. Are there any errors or assumptions that may have affected the result? Carefully verify each operation."*

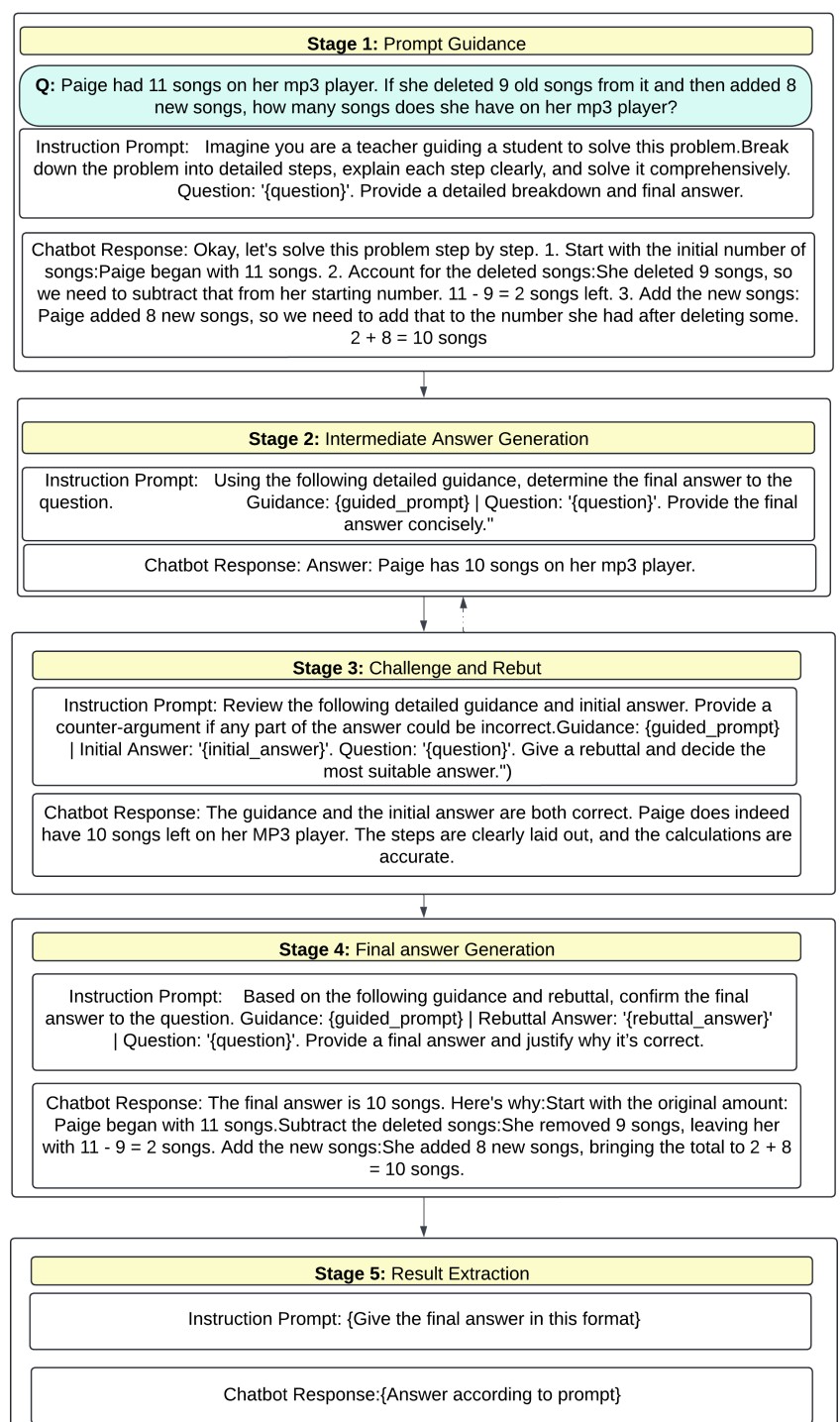

Figure 1: Adaptive Prompting Framework

During this phase, the model detects potential issues. For instance, it might note that while selling a quarter of the apples ($26/4 = 6.5$) is mathematically correct, it is unrealistic in a real-world context since apples cannot be sold in fractional quantities. The model adjusts the calculation to account for integer rounding, concluding that the farmer sells 6 apples instead of 6.5. The revised reasoning becomes:

> *"The farmer sells $26/4 = 6$ apples (rounding down to account for whole apples), leaving $26 - 6 = 20$ apples."*

This iterative process introduces a higher degree of reliability, as errors and unrealistic assumptions are dynamically corrected. Unlike CoT, where errors propagate unchecked, **Adaptive Prompting** incorporates mechanisms for continual improvement, ensuring a robust and logically sound output.

The final phase consolidates the validated steps into a well-justified solution, presented in a clear and actionable format. For the given example, the model confidently concludes:

> *"After accounting for the corrections, the farmer has 20 apples in total."*

This multistage process ensures that the reasoning is not only accurate, but also reflective of real-world constraints, enhancing the utility of the model in practical applications.

Beyond solving arithmetic problems, Adaptive Prompting can generalize to other domains, such as scientific reasoning, legal analysis, and complex decision-making tasks. The structured methodology is particularly beneficial for smaller language models, enabling them to perform competitively with larger models while maintaining computational efficiency. By emphasizing understanding, iterative refinement, and error correction, **Adaptive Prompting** represents a significant advancement over static prompting techniques, aligning AI reasoning processes more closely with human cognition.

## 5 RESULTS AND DISCUSSION

The results of our experiments across multiple datasets—SVAMP, GSM8K, AddSub, MultiArith, SingleEq, and AQuA—demonstrate a compelling case for the efficacy of dynamic prompting over traditional zero-shot approaches. We evaluated both GPT-3.5 and GPT-4 using zero-shot and dynamic prompting methods, and the accuracy across all datasets highlights the advantages of this approach.

Table 1: Accuracy on ten datasets from three categories of reasoning tasks.

| Model | MultiArith | GSM8K | AddSub | AQUA | SingleEq | SVAMP | CSQA | Strategy |
|---|---|---|---|---|---|---|---|---|
| **GPT-3.5 Turbo** | | | | | | | | |
| Zero-Shot-CoT | 95.3 | 78.9 | 85.8 | 53.0 | 93.5 | 79.3 | 72.3 | 66.1 |
| Few-Shot-CoT | 97.8 | 82.3 | 92.1 | 60.2 | 94.9 | 82.5 | 74.5 | 68.5 |
| **GPT-4** | | | | | | | | |
| Zero-Shot-CoT | 97.8 | 94.6 | 92.4 | 72.8 | 95.0 | 90.4 | - | - |
| Few-Shot-CoT | 98.1 | 97.1 | 95.1 | **77.1** | 96.0 | **94.2** | - | - |
| **Gemma 9B** | | | | | | | | |
| Adaptive Prompting | **99.44** | **98.72** | **96.37** | **77.1** | **99.4** | **93.0** | **94.0** | **82.0** |
| Zero-Shot-CoT | 92.0 | 60.5 | 75.0 | 40.5 | 85.0 | 65.0 | 67.0 | 55.0 |
| Few-Shot-CoT | 95.5 | 68.6 | 85.5 | 52.0 | 89.0 | 72.5 | 70.0 | 61.0 |

The results of **Gemma 9B** on a variety of reasoning tasks, including arithmetic problem-solving, word problems, and commonsense reasoning, demonstrate the model's superior performance with **Adaptive Prompting** when compared to traditional zero-shot and few-shot approaches. In the arithmetic reasoning tasks, **Gemma 9B** achieves exceptional accuracy with **Adaptive Prompting**, with scores of **99.44%** on **MultiArith**, **96.37%** on **AddSub**, and **99.4%** on **SingleEq**. These results significantly exceed those of **GPT-3.5 Turbo** and **GPT-4**, which show lower accuracy across these datasets. The improvement observed in these tasks suggests that **Gemma 9B**'s adaptive prompting method is highly effective in leveraging task-specific context to solve arithmetic reasoning problems. In comparison, when evaluated in the zero-shot setting, **Gemma 9B**'s performance drops, with accuracies of **92.0%**, **75.0%**, and **85.0%** on **MultiArith**, **AddSub**, and **SingleEq**, respectively. This indicates that while **Gemma 9B** performs well in zero-shot tasks, its performance significantly improves when prompted with task-specific information, highlighting the benefits of adaptive prompting over traditional zero-shot inference.

For tasks involving more complex multi-step reasoning, such as **GSM8K** and **SVAMP**, **Gemma 9B** again outperforms **GPT-4** and **GPT-3.5 Turbo** when adaptive prompting is employed. With **Adaptive Prompting**, **Gemma 9B** achieves **98.72%** accuracy on **GSM8K** and **93.0%** on **SVAMP**, compared to **GPT-4**'s few-shot performance of **97.1%** on **GSM8K** and **94.2%** on **SVAMP**. These results show that while **GPT-4** performs well on these tasks, **Gemma 9B**'s adaptive prompting further enhances its performance. However, in the few-shot setting, **Gemma 9B**'s performance on **GSM8K** drops to **68.6%** Team et al. (2024), underscoring the importance of providing context in order to achieve better results on multi-step reasoning tasks.

In the commonsense reasoning tasks, such as **CSQA** and **StrategyQA**, **Gemma 9B** excels with **Adaptive Prompting**, achieving accuracies of **94.0%** and **82.0%**, respectively. These results surpass both **GPT-3.5 Turbo** and **GPT-4** in their few-shot settings. Notably, **Gemma 9B**'s adaptive prompting technique enables it to generate more accurate commonsense inferences, a domain in which traditional few-shot methods often struggle. On the other hand, **Gemma 9B**'s zero-shot performance on **CSQA** and **StrategyQA** is relatively lower, with accuracies of **67.0%** and **55.0%**, respectively, highlighting the limitations of zero-shot reasoning in complex commonsense tasks.

The results from this experiment provide strong evidence that adaptive prompting can empower smaller language models to perform at competitive levels with much larger models, such as **GPT-4**. While large models like **GPT-4** exhibit impressive performance, especially in few-shot settings, our adaptive prompting framework enables smaller models to achieve comparable results without the need for fine-tuning or task-specific data. By dynamically adjusting prompt structures and incorporating real-time validation and self-correction, our approach enhances the reasoning capabilities of these smaller models, improving their accuracy in diverse tasks like arithmetic and logical reasoning, as well as commonsense reasoning. This finding underscores the importance of dynamic, iterative prompting techniques, which not only improve model performance but also maintain computational efficiency. Overall, our work highlights the untapped potential of smaller models when paired with advanced prompting strategies, offering a promising direction for future research in efficient, scalable natural language processing systems.

## 6 FUTURE WORK

While our research demonstrates significant progress in enhancing the reasoning capabilities of smaller language models, several limitations and challenges necessitate further exploration. One major issue lies in the variability of effectiveness across different domains and tasks. Although the multi-stage reasoning framework provides distinct advantages, its adaptability often demands task-specific calibration. Future research should focus on establishing standardized assessment metrics and methodologies to ensure consistent performance across diverse applications, minimizing the need for extensive manual tuning.

Ethical considerations are paramount as the methodology evolves. The iterative and reflective nature of the framework, while a strength in fostering deeper reasoning, also introduces risks of amplifying biases or propagating harmful patterns embedded in the training data. Minor errors in intermediate steps can compound over multiple reasoning stages, potentially leading to undesirable outcomes. To mitigate these risks, future work should prioritize integrating robust safeguards for bias detection and correction at every stage of the reasoning process. These safeguards must include mechanisms to flag and address problematic outputs during both training and deployment, ensuring that the iterative refinement process does not inadvertently reinforce existing biases.

Moreover, this multi-stage reasoning framework offers significant potential for enhancing AI's security against adversarial attacks. Adversarial prompts, which are intentionally designed to exploit vulnerabilities or trigger harmful outputs, often embed complex or hidden nuances that can bypass traditional defenses. By systematically breaking down prompts into smaller, interpretable steps, the framework can uncover these hidden elements and analyze their implications. For example, when a prompt includes subtle manipulations designed to elicit unsafe or unintended outputs, the structured reasoning process can identify and isolate such patterns, effectively neutralizing the attack.

The framework also enables proactive safeguards by blocking harmful responses before they are fully generated. If any stage of the reasoning process detects unsafe or undesirable information, the model can halt further output generation, issue a warning, or request clarification. This layered

approach enhances the robustness of AI systems by ensuring that potentially harmful prompts are addressed dynamically, rather than relying solely on static filters or pre-defined safeguards.

Developing transparent and ethical guidelines for the implementation and deployment of such security-enhanced frameworks is essential. This includes establishing best practices for detecting adversarial behavior, defining clear thresholds for harmful content, and ensuring that safeguards do not disproportionately affect legitimate uses. By embedding these principles into the design, this methodology not only advances the robustness and reliability of AI systems but also fosters trust and accountability in their deployment.

Lastly, exploring collaborative applications of this methodology in human-AI systems presents an exciting frontier. Integrating adaptive models with human feedback loops in domains such as education, decision-making, or healthcare could combine human intuition with AI's scalability and consistency. Such systems could facilitate robust problem-solving while maintaining alignment with human values and expectations.

## 7 CONCLUSION

Adaptive Prompting provides a structured approach to improving reasoning in language models by combining task decomposition, iterative refinement, and self-reflection. While it shows promise in enabling smaller models to perform competitively in reasoning tasks, its contributions should be viewed as complementary to existing techniques rather than a replacement. This framework demonstrates the value of optimizing reasoning processes rather than solely increasing model size, offering insights into efficient task interaction for AI applications.

Future work will focus on broadening the evaluation of Adaptive Prompting, exploring its performance across different models, datasets, and real-world scenarios. By addressing these questions, we aim to further establish Adaptive Prompting as a scalable and reliable framework for reasoning-intensive tasks.

## 8 ACKNOWLEDGMENTS

I would like to extend my sincere thanks to the anonymous peer reviewers for their valuable feedback and suggestions, which have significantly improved the quality of this paper.

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
