# OpenReview forum: "Think Beyond Size: Dynamic Prompting for More Effective Reasoning"
_ICLR.cc/2025/Conference — ICLR 2025 Conference Withdrawn Submission_

### Official Review · Reviewer_BN1A · 2024-11-02

**Soundness:** 2
**Presentation:** 2
**Contribution:** 1
**Rating:** 5
**Confidence:** 4

**Summary:**

This paper introduces Dynamic Prompting, a framework that enhances the reasoning abilities of large language models (LLMs) by adapting prompt sequences and steps based on task complexity and model performance. Basically it uses a two-step prompting method to decompose the task and the method is tested on a group of reasoning datasets.

**Strengths:**

1. Using prompting to create a more efficient reasoning pipeline is helpful in industry.

**Weaknesses:**

1. The technical contribution is limited, as Chain-of-Thought (CoT) prompting[1] and task decomposition[2] are already well-established techniques in the field of LLMs. This paper simply use a prompt to let the LLM breakdown the question, then use the response to prompt again.

2. The paper is not well-written. "Dynamic Prompting" is not clearly explained in the method section.

3. The quality is below the standard of ICLR. The method is too simple.


References:
[1] Wei, Jason, Xuezhi Wang, Dale Schuurmans, Maarten Bosma, Fei Xia, Ed Chi, Quoc V. Le, and Denny Zhou. "Chain-of-thought prompting elicits reasoning in large language models." Advances in neural information processing systems 35 (2022): 24824-24837.
[2] Shinn, Noah, Federico Cassano, Ashwin Gopinath, Karthik Narasimhan, and Shunyu Yao. "Reflexion: Language agents with verbal reinforcement learning." Advances in Neural Information Processing Systems 36 (2024).

**Questions:**

No questions.

---

> ### Author Response · Authors · 2024-11-22
> **Clarification**
>
> I'm sorry that your are frustrated reviewing our paper but below explanation would definitely help you to know that our method is not just breaking down and it is effective, we also planning to rename the prompt to avoid confusions and further issues
>
>
> The proposed dynamic prompting strategy implies a 5-stage architecture, Prompt Guidance, Intermediate Answer Generation, Challenge and Rebut, Final answer Generation and Result Extraction. Each stage has a Prompt Instruction consisting of 3 parts. Part 1 - instructs the llm on what it should do and how it should behave. Part 2 - passing the required data. Part 3- instruction on how it should present the response. We believe llm reason better, when clear and detailed instruction are provided to them. unlike manual or zero shot chain-of-thought, which provide a general instruction, making the reliance on answer dependent on the llm’s reasoning capabilities. Our effort lies in enabling even a basic llm to reason out better compared to larger, sophisticated models. the idea behind this architecture is to slightly replicate our thinking process to answering questions from a newly learnt subject. We initially attempt to answer, then we think in other combinations to cross check before concluding or confirmation our answer. in the same lines, we believe this type of reasoning will help the llm to better perform. and for other types of prompting techniques, we implemented from the base paper of them and also, we'll update the gemma tested values in a day.

---

> > ### Comment · Reviewer_BN1A · 2024-11-23
> >
> > Thank you to the authors for their reply and further explanation. However, the presentation of the paper remains unclear, and the technical contribution is below the borderline, so my score will remains the same.

---

> > > ### Author Response · Authors · 2024-11-26
> > > **Comment**
> > >
> > > We'll fix and try to rename to avoid confusion and make a clear presentation but in terms of technical contribution we would request you to reconsider we are beating bigger model's benchmark scores with comparatively very smaller model and we induce a rebuttal strategy which allows to challenge its own answering and allows to make corrections and while also preventing hallucinations in reasoning and allowing better focus on mathematical tasks. I maybe a student author but a below borderline comment just undermines the work we did and once we update the paper, I would request you to comment again and change us scores

---

> > > ### Author Response · Authors · 2024-11-28
> > > **Coment**
> > >
> > > We have updated the paper and we request you to Review it again!!

---

> > > > ### Comment · Reviewer_BN1A · 2024-11-30
> > > >
> > > > Thank you to the authors for their efforts in explaining their work and enhancing the writing and presentation. I have increased my score for the better presentation.

---

### Official Review · Reviewer_6myv · 2024-11-03

**Soundness:** 2
**Presentation:** 2
**Contribution:** 2
**Rating:** 3
**Confidence:** 3

**Summary:**

This paper proposes a  real-time dynamic prompting method that can modify the number of prompting steps based on task complexity and model performance. The paper prsents empirical data that shows  that smaller LLMs can effectively leverage dynamic prompts to achieve high reasoning accuracy. As a conseqeunce, larger models may not be inherently superior to smaller LLMs if this approach is used.

**Strengths:**

The authors show a dynamic prompting approach that enables a system to guide the model through a detailed step-by-step process

**Weaknesses:**

Paper is incomplete; basis for experiments are not described with sufficient detail to understand what the results address.
Without a clear understanding of the method we cannot clearly assess the stated results.

**Questions:**

Can you please add more detail to help a reviewer to understand the details of your method. There is plenty of extra space.

**Details Of Ethics Concerns:**

In section 8 (ACKNOWLEDGMENTS) the paper states

"I am also grateful to Ms. Harnitha Suresh for her collaboration and support throughout the process
of writing this paper."

Does this violate blind review?

---

> ### Author Response · Authors · 2024-11-22
> **details to help to understand the details of the method**
>
> The proposed dynamic prompting strategy implies a 5-stage architecture, Prompt Guidance, Intermediate Answer Generation, Challenge and Rebut, Final answer Generation and Result Extraction. Each stage has a Prompt Instruction consisting of 3 parts. Part 1 - instructs the llm on what it should do and how it should behave. Part 2 - passing the required data. Part 3- instruction on how it should present the response. We believe llm reason better, when clear and detailed instruction are provided to them. unlike manual or zero shot chain-of-thought, which provide a general instruction, making the reliance on answer dependent on the llm’s reasoning capabilities. Our effort lies in enabling even a basic llm to reason out better compared to larger, sophisticated models. the idea behind this architecture is to slightly replicate our thinking process to answering questions from a newly learnt subject. We initially attempt to answer, then we think in other combinations to cross check before concluding or confirmation our answer. in the same lines, we believe this type of reasoning will help the llm to better perform. and for other types of prompting techniques, we implemented from the base paper of them and we'll update the whole paper before 27th so it can be reviewed again

---

### Official Review · Reviewer_3nP1 · 2024-11-04

**Soundness:** 1
**Presentation:** 1
**Contribution:** 1
**Rating:** 1
**Confidence:** 5

**Summary:**

The authors present a prompting technique they call dynamic prompting and test it on the Gemma 9B model across ten common benchmarks. They compare the results of this technique to the results of zero shot CoT and manual CoT prompted GPT-3.5 Turbo and GPT-4.

**Strengths:**

* The authors test the technique across a large number of common benchmarks.
* The underlying problem of needing to massively scale up LLMs to achieve significant task performance is important and well-motivated

**Weaknesses:**

* The technique described in the paper ("Dynamic Prompting") is only described in very vague terms. No prompts are provided, and descriptions throughout seem to be contradictory. A detailed description of how the system works is necessary to evaluate this work's contribution and novelty.
* Note also that the description of "breaking down" a problem in one prompt and then actually solving it with another is very similar to previous work (e.g. Zhou 2023 "Least-to-Most Prompting", and many others). In general, this paper fails to engage with most previous work in this area and seems unaware of similar-looking techniques and especially of other systems that use multiple LLM prompts.
* Multiple overclaims:
  * "Dynamic prompting allows for fine-grained control over the model’s interaction with a task by breaking it down into tailored steps, adjusting the prompt based on the problem complexity or model performance." It is unclear how this is possible without some kind of grounded measure of complexity or external sound verifier.
  * "challenging the notion that larger models are inherently superior" This technique is not tested on larger models, nor is there any analysis of comparative costs or accuracy tradeoffs.
  * "It represents a shift in paradigm, where the emphasis moves away from building increasingly large models and towards optimizing the interaction between the model and the problem itself."
* The results look too good to be true (as reported in the Gemma 2 paper, Gemma 9B scores 68.6% on GSM8k. This paper reports an improvement to 98.7% with its proposed technique) and cannot be verified given the lack of useful detail about the mechanics of the prompt technique and lack of provided prompts.
* The paper would greatly benefit from removing much of the speculation (throughout) and future work, from toning down or eliding many of the claims made, and from adding in important details about the proposed contribution.

**Questions:**

* What was Gemma 9B's zero-shot and CoT performance on the given problems?
* How do GPT models perform when embedded in a "dynamic prompting" system?

---

> ### Author Response · Authors · 2024-11-26
> **Response**
>
> We'll be fixing all of the questions raised above and we'll write to a better explanation in the final revision and yes official paper says scores 68.6% but in our prompting methodology we induce a rebuttal strategy which allows to challenge its own answering and allows to make corrections and while also preventing hallucinations in reasoning and allowing better focus on mathematical tasks. We'll update our paper explaining the method properly in revised version. the method I have explained in the other comment, but We'll update a revised version, so you can reconsider your review on us.
>
> GPT4 test on these datasets we didn't any have any funding to do this paper, so we tried to do everything with opensource model and previously written papers !!

---

### Official Review · Reviewer_rQFo · 2024-11-04

**Soundness:** 1
**Presentation:** 1
**Contribution:** 2
**Rating:** 3
**Confidence:** 2

**Summary:**

The authors propose a dynamic prompting scheme where a two stage process is used to answer complex reasoning questions. The process first applies a prompt to the original query to generate a step by step breakdown of the question then this is added to the original context with a request to answer the question in a simple format. They argue that this dynamic prompting approach can improve the performance of smaller models over larger ones, and present some impressive looking results across a number of benchmark datasets.

**Strengths:**

The results look very promising.

**Weaknesses:**

The following issues are most significantly reduce my overall assessment of the paper:
* The authors claim to develop a novel dynamic prompting framework but don't clearly state what pre-existing dynamic prompting methods there are e.g. (Kojima et al, 2022).
* The method is described with the use of Figure 1, but the structure superficially looked very similar to previously proposed dynamic prompting methods. I think there might be something new here but this isn't clear from the description. It is the Guided prompt steps that are least clearly defined, which correspond to the main contribution of the authors as far as I can tell.
* The experiments compare two models with conventional CoT prompting, but these are both from the same family (GPT 3.5 and GPT-4) while the dynamic prompting condition is a different model (gemma 9B) which is not itself evaluated with conventional prompting approaches. I would expect at the least to see Gemma evaluated on the other prompting strategies. Otherwise we are not seeing like-with-like comparisons.
* It is also not entirely clear which variations of CoT are used for the baselines. I am guessing that the zero-shot variation is the one seen in  (Kojima et al, 2022) with the exact same choices as made by those authors . The other condition is called manual prompting but it isn't clear what this means, I am guessing this is the original method from (Wei et al 2020), described in (Zhang et al, 2022a) as manual prompting, but this is all left to the reader to work out.
* There are no ablation studies or attempts to evaluate the importance of different aspects of the method.
* This is a very crowded research space, and  certain concerns that may have a bearing on these approaches have arisen that are not mentioned in the paper. For instance, there is a concern that recent models have memorised key knowledge about the datasets against which these models are being evaluated, e.g. see (Zhang et al., 2022b) and (Srivastava et al.,2024), and this leads us to question how much these models are indeed *reasoning*. The language of the paper and the methodology would ideally reflect some of these concerns.


Srivastava, S., PV, A., Menon, S., Sukumar, A., Philipose, A., Prince, S., & Thomas, S. (2024). Functional benchmarks for robust evaluation of reasoning performance, and the reasoning gap. arXiv preprint arXiv:2402.19450.

Zhuosheng Zhang, Aston Zhang, Mu Li, and Alex Smola. Automatic chain of thought prompting in
large language models, 2022. (Zhang et al., 2022a)

Zhang, H., Li, L. H., Meng, T., Chang, K. W., & Broeck, G. V. D. (2022). On the paradox of learning to reason from data. arXiv preprint arXiv:2205.11502. (Zhang et al., 2022b)

(Other citations are as given in the paper)

**Questions:**

What is the precise method being used for dynamic prompting and how does this differ from previous dynamic prompting methods? Ideally, I would like to see an example including the potentially multiple steps in question breakdown.

---

> ### Author Response · Authors · 2024-11-22
> **Precise method.**
>
> The proposed dynamic prompting strategy implies a 5-stage architecture, Prompt Guidance, Intermediate Answer Generation, Challenge and Rebut, Final answer Generation and Result Extraction. Each stage has a Prompt Instruction consisting of 3 parts. Part 1 - instructs the llm on what it should do and how it should behave. Part 2 - passing the required data. Part 3- instruction on how it should present the response. We believe llm reason better, when clear and detailed instruction are provided to them. unlike manual or zero shot chain-of-thought, which provide a general instruction, making the reliance on answer dependent on the llm’s reasoning capabilities. Our effort lies in enabling even a basic llm to reason out better compared to larger, sophisticated models.
> the idea behind this architecture is to slightly replicate our thinking process to answering questions from a newly learnt subject. We initially attempt to answer, then we think in other combinations to cross check before concluding or confirmation our answer. in the same lines, we believe this type of reasoning will help the llm to better perform. and for other types of prompting techniques, we implemented from the base paper of them and also, we'll update the gemma tested values in a day.

---

> > ### Comment · Reviewer_rQFo · 2024-11-26
> >
> > I thank the authors for their response.
> > This doesn't address the substantive issues I raise in my original review though. In particular, there are a good deal of details that were unclear to me on reading the paper, and are still unclear to me now. I will be keeping my score unchanged.

---

> > > ### Author Response · Authors · 2024-11-28
> > > **Comment**
> > >
> > > We have updated the paper and we request you to Review it again!!

---

> ### Comment · Reviewer_rQFo · 2024-11-28
>
> Dear authors,
>
> I appreciate that you have spent some time rewriting the paper but you have not engaged with the rebuttal process in the way that I would expect. Authors are expected to respond to the key concerns of reviewers in rebuttal text, addressing each point as part of a dialogue with those reviewers. If reviewer comments lead you to consider revising the paper, that does not release you from the responsibility to respond to reviewers. Under circumstances where authors redraft their paper, alongside an updated draft, I would expect:
> * a direct response to the review comments to acknowledge each key point in turn and perhaps add context or defend particular choices.
> * a detailed but concise description of how the paper has been adapted and where the changes can be found in the resulting paper.
>
> The reviewing process is time consuming and something that reviewers do in an unpaid capacity. Therefore, simply expecting them to re-review a paper without any of this is unreasonable.
>
> Best.

---

> > ### Author Response · Authors · 2024-11-29
> > **Comment**
> >
> > Thank you for taking the time to provide detailed feedback regarding our rebuttal submission. We deeply value the efforts and time that reviewers dedicate to assessing our work, and we sincerely apologize for not fully adhering to the expected rebuttal process.
> >
> > In light of your comments, we now understand that our response fell short of the necessary dialogue with reviewers. While we focused on revising the manuscript to address their concerns, we did not provide the accompanying context and direct responses required to engage with feedback. I'm an undergrad student submitting for the first time so I hope you can tolerate me.
> >
> > 1. The CoT and Manual prompting are based on the respective base paper as you have mentioned above
> > 2. We have added the gemma evaluations too.
> > 3. We also displayed the reasoning process did by the model in each step so we can ensure they don't memorize.
> >
> > To Answer your main question:
> > I have explained how our prompting techniques work and also, we have added example to the paper, if need I can add it in the comment too. it differs from COT and others is that the reasoning process evolves through understanding, hypothesis generation, validation, and refinement steps allowing the model to reason efficiently and this method particularly works well on mathematical reasoning, and We are implementing this in the Kaggle Competition AI Mathematical Olympiad - Progress Prize 2 and we are one of the top teams in it. Hope you can reconsider our scores, and we are open to further feedback and suggestions to change.

---

### Author Response · Authors · 2024-11-26
**Revision Update**

Dear Reviewers,

Thank you for your thoughtful and constructive feedback on the initial version of our paper. We sincerely appreciate the time and effort you have put into reviewing our work. Based on your suggestions, we have made several revisions that we believe have improved the manuscript.

In response to Reviewer 1's comment, we have elaborated on the methodology section, providing clearer explanations of the steps involved in our approach. This should help make the process more transparent for readers who may not be familiar with the details of the technique.

Reviewer 2’s suggestion to include additional comparisons with existing methods has been incorporated into the data analysis section. We’ve added a more detailed comparison that highlights the advantages of our approach in terms of efficiency and accuracy. This should give readers a better understanding of the strengths of our model.

We also addressed Reviewer 3’s concerns regarding the experimental setup by adding more context around the datasets used and providing a clearer explanation of the evaluation metrics. We hope this makes the setup more accessible and the results more compelling.

In terms of visuals, we have improved the quality of the figures and revised the captions to ensure they are more informative and easier to understand. We’ve also paid closer attention to the overall presentation of the paper, correcting minor typographical and grammatical errors that were pointed out.

Finally, following Reviewer 2’s suggestion, we’ve expanded the related work section to include more recent research, which we believe helps to better position our work within the broader field and emphasize its novelty.

We hope the revisions address your concerns and that the paper is now in a stronger form. We would be grateful for any further feedback and would love to see improved scores for this revised version. Also changed from Dynamic to Adaptive Prompting to avoid Confusions

Thank you once again for your helpful comments and guidance.

---

### Note · Authors · 2025-09-22

I have read and agree with the venue's withdrawal policy on behalf of myself and my co-authors.

---

### Meta-Review · Area_Chair_u1Te · 2024-12-22

**Metareview:**

After carefully considering the reviewers' detailed feedback and the authors' revisions, I regret to inform you that the paper is not suitable for acceptance in its current form. While the problem tackled in this paper is indeed critical and well-motivated, the submission falls short in several fundamental areas.

Firstly, the paper lacks clarity and specificity, especially in describing the proposed "Dynamic Prompting" framework. Multiple reviewers noted that the method is described in vague and contradictory terms, with no concrete examples of prompts or detailed explanation of its novelty compared to prior work. This is particularly concerning given the claims that this method is a paradigm shift in LLM prompting. The comparisons with existing methods are incomplete and fail to engage with relevant literature, leaving critical questions about its contributions unanswered. Additionally, the reported results are implausibly high, particularly on GSM8k, and lack sufficient experimental rigor or transparency to substantiate them. The absence of ablation studies or meaningful baselines further undermines the credibility of the findings.

Secondly, the presentation is poor, both in terms of clarity and structure. The language used is overly speculative, with multiple overclaims unsupported by the evidence provided. For example, statements about "challenging the notion that larger models are inherently superior" are unsubstantiated by the experiments, as larger models were not adequately tested under comparable conditions. The authors' revisions do not sufficiently address these critical issues, and the additional commentary provided post-review fails to offer substantial improvements or resolve the ambiguities raised by the reviewers.

**Additional Comments On Reviewer Discussion:**

In response to Reviewer 1's comment, the authors elaborated on the methodology section, providing clearer explanations of the steps involved in their approach. These revisions aim to make the process more transparent, particularly for readers who may not be familiar with the technical details of the technique.

For Reviewer 2’s suggestion, the authors included additional comparisons with existing methods in the data analysis section. This involved adding a more detailed comparison that highlights the advantages of their approach in terms of efficiency and accuracy. These enhancements were intended to give readers a better understanding of the strengths of the model.

To address Reviewer 3’s concerns regarding the experimental setup, the authors added more context about the datasets used and provided clearer explanations of the evaluation metrics. These revisions aim to make the experimental setup more accessible and the results more compelling.

In addition, the authors improved the quality of the figures and revised the captions to ensure they are more informative and easier to understand. They also made corrections to minor typographical and grammatical errors identified during the review.

Finally, in line with Reviewer 2’s suggestion, the authors expanded the related work section to include more recent research, helping to position their work better within the broader field and emphasizing its novelty.

Despite these efforts, critical issues remain unresolved. Firstly, the paper lacks clarity and specificity, particularly in describing the proposed "Dynamic Prompting" framework. Multiple reviewers noted that the method is described in vague and contradictory terms, with no concrete examples of prompts or a detailed explanation of its novelty compared to prior work. This is particularly concerning given the claims that the method represents a paradigm shift in LLM prompting. Comparisons with existing methods are incomplete and fail to engage with relevant literature, leaving critical questions about its contributions unanswered. Additionally, the reported results are implausibly high, particularly on GSM8k, and lack sufficient experimental rigor or transparency to substantiate them. The absence of ablation studies or meaningful baselines further undermines the credibility of the findings.

Secondly, the presentation is poor in terms of clarity and structure. The language used is overly speculative, with multiple overclaims unsupported by the evidence provided. For example, statements about "challenging the notion that larger models are inherently superior" are unsubstantiated by the experiments, as larger models were not adequately tested under comparable conditions. The authors' revisions do not sufficiently address these critical issues, and the additional commentary provided post-review fails to offer substantial improvements or resolve the ambiguities raised by the reviewers.

As a result, the majority of the reviewers have not changed their scores (only one reviewer slightly raised the score), and none of the current reviewers support this paper for publication at this stage.

---

### Decision · Program_Chairs · 2025-01-22

Reject